# NMR, LC-MS Characterization of *Rydingia michauxii* Extracts, Identification of Natural Products Acting as Modulators of LDLR and PCSK9

**DOI:** 10.3390/molecules27072256

**Published:** 2022-03-30

**Authors:** Stefania Sut, Aminallah Tahmasebi, Nicola Ferri, Irene Ferrarese, Ilaria Rossi, Giovanni Panighel, Maria Giovanna Lupo, Filippo Maggi, Akbar Karami, Stefano Dall’Acqua

**Affiliations:** 1Department of Pharmaceutical and Pharmacological Sciences, University of Padova, 35122 Padova, Italy; stefania.sut@unipd.it (S.S.); irene.ferrarese@unipd.it (I.F.); ilaria.rossi.11@studenti.unipd.it (I.R.); giovanni.panighel@studenti.unipd.it (G.P.); mariagiovanna.lupo@unipd.it (M.G.L.); 2Department of Agriculture, Minab Higher Education Center, University of Hormozgan, Bandar Abbas 79177, Iran; tahmasebi.info@yahoo.com; 3Plant Protection Research Group, University of Hormozgan, Bandar Abbas 79177, Iran; 4Department of Medicine, University of Padova, 35122 Padova, Italy; nicola.ferri@unipd.it; 5School of Pharmacy, University of Camerino, 62032 Camerino, Italy; filippo.maggi@unicam.it; 6Department of Horticultural Science, School of Agriculture, Shiraz University, Shiraz 7134754331, Iran

**Keywords:** *Rydingia michauxii*, PCSK9, LDL receptor, LC-DAD-MS, NMR, phenolic constituents, labdane, iridoids

## Abstract

*Rydingia michauxii* (Briq.) Scheen and V.A.Albert (Lamiaceae) is used in Iranian traditional medicine to treat malaria, diabetes, hyperlipidemia, rheumatism and cardiovascular diseases. NMR and LC-DAD-MS^n^ analyses were used to establish extract composition and phenylethanoid, flavonoid glycosides, lignans, labdane diterpenes and iridoids were identified and quantified. The main constituents were isolated, and structures were elucidated based on NMR, polarimetric and MS measurements. A new natural compound, ent-labda-8(17),13-dien-18-glucopyranosyl ester-15,16-olide is described here. The effects of ent-labda-8(17),13-dien-18-oic acid-15,16-olide (**1**), ent-labda-8(17),13-dien-18-glucopyranosyl es-ter-15,16-olide (**2**), antirrhinoside (**3**), echinacoside (**4**), verbascoside (**5**), and apigenin 6,8-di-C-glucoside (**6**), on the low-density lipoprotein receptor (LDLR) and proprotein convertase subtilisin/kexin type 9 (PCSK9), were studied in the human hepatocarcinoma cell line Huh7. Among the six constituents, (**3**) showed the strongest induction of the LDLR (3.7 ± 2.2 fold vs. control) and PCSK9 (3.2 ± 1.5 fold vs. control) at a concentration of 50 µM. The in vitro observations indicated a potential lipid lowering activity of (**3**) with a statin-like mechanism of action.

## 1. Introduction

The genus *Rydingia* (syn. Otostegia, Lamiaceae family) comprises about thirty three species distributed in Asia and northeastern Africa. *Rydingia michauxii* (Briq.) Scheen and V.A.Albert is endemic to subtropical areas of Fars province, Iran, and is one of the four species of the genus *Rydingia* (*R. integrifolia* (Benth.) Scheen and V.A.Albert, *R. limbata* (Benth.) Scheen and V.A.Albert, and *R. persica* (Burm.f.) Scheen and V.A.Albert) that share some distinct morphological characters like spines at the leaf axils as well as spinose and persistent bracteoles, yellow flowers, and few-flowered verticillaster [1,2,3,4]. Labdanes are the main constituents of this genus, followed by abietane, pimarane, hispanane, and clerodane diterpenoids. Several other metabolites have been identified in the genus *Rydingia* in particular flavonoids, triterpenoids, steroids, carboxylic acids, carotenoids, nitrogen containing compounds and phenylpropanoids [5]. Recently labdane constituents have been identified from *R. persica* and were evaluated for their anti-inflammatory activity [5,6].

Only limited information on *R. michauxii* is reported in the literature, mainly focusing on the essential oil characterization. Previous work identified caryophyllene oxide (20.1%), trans-verbenol (10.2%), linalool (5.3%) and humulene epoxide II (4.6%) as major constituents [2]. Investigation of essential oil composition of *R. michauxii* was evaluated during the dormant, vegetative, and flowering stages and showed that chemical compounds were altered during the various developmental stages [4]. Results indicated that chemical constituents of *R. michauxii* at the dormant stage were eugenol (36.81%), eugenol acetate (21.02%), and carvacrol (9.35%), while variation in chemical compositions of the plant species were observed during the various developmental stages [4]. *Rydingia* species such as *R. persica,* are used in traditional medicine in Iran for diabetes, rheumatism, cardiac distress, gastric discomfort, hypertension, cold, hyperlipidemia, headache, reducing palpitations, and as a laxative, carminative, antipyretic, parasite repellent, sedative, and an addiction treatment [4]. Notably, *R. michauxii* is used in the treatment of malaria [5].

Statins represent a milestone in the treatment of hypercholesterolemia, with highly documented effectiveness in reducing cardiovascular (CV) endpoints and managing low-density lipoprotein-cholesterol (LDL-C) levels [7]. Of importance, robust meta-analyses suggest that the relative benefit of LDL-C reduction will be greater in primary rather than secondary prevention when a greater atherosclerotic plaque burden is already established [8,9]. Despite their proven efficacy in preventing cardiovascular disease, the adherence to statin therapy and the levels of LDL-C in the European population is far below the optimal range [10]. Several reasons could explain the failure of statins to achieve the recommended LDL-C goal, including the development of side effects, such as statin induced myopathy [11]. It therefore becomes increasingly urgent to find new approaches to efficiently reduce LDL-C levels in primary prevention. Encouraging a healthy lifestyle, tackling poor health habits, and reducing CV risk factors such as the aforementioned LDL cholesterol, should be encouraged. The pivotal role of nutrition in the prevention of CV disease has been widely investigated and innovative nutritional strategies to reduce the main CV risk factors have been developed. There is an interest in natural products in this field and nowadays one of the most used lipid-lowering nutraceuticals is represented by red yeast rice (RYR) that is a natural source of statins [12]. However, concerns regarding the safety of RYR have been raised after the publication of some case reports claiming toxicity [13,14,15]. From these considerations, the identification of additional natural compounds that may be used in the current armamentarium for controlling LDL-C levels in the general population becomes relevant.

In this study we considered *R. michauxii* starting from its traditional use related to hypertension, cardiac distress, and hyperlipidemia as a potential source of natural cholesterol controlling agents. At first, we proceeded with chemical investigation of the leaf extracts of *R. michauxii*. NMR and HPLC-DAD-MS^n^ were selected as analytical approaches that allowed investigation of this class of constituents. Furthermore, quantitative investigations allowed comparisons in the dormant, vegetative and flowering stages. The observation that the vegetative and flowering stages were most abundant in secondary metabolites, selected these extracts for the isolation of the most abundant derivatives namely phenolics and terpenoids. The isolated compounds were evaluated as cholesterol lowering agents in an in vitro model. To explore a possible hypocholesterolemic effect of the isolated compounds, we evaluated their effects on the expression of low-density lipoprotein receptor (LDLR) and proprotein convertase subtilisin/kexin type 9 (PCSK9) in the human hepatocarcinoma cell line Huh7. Furthermore, the levels of cholesterol in the same cell lines after treatment with the most promising compounds were measured.

## 2. Results

### 2.1. Phytochemical Characterization of the Extracts

To assess the constituents of the plant extracts different approaches were combined. NMR analysis can be a powerful tool to assess preliminary composition of complex mixtures with a “metabolomic” approach and previously published papers support the usefulness of 1D and 2D NMR of complex matrix-like plant extracts in the identification of compounds and also for quantification purposes in several applications [16,17,18]. In our lab we applied this approach to different plant extracts to detect various classes of constituents by NMR and combined those data with the data obtained with liquid chromatography coupled with mass spectrometry [19,20,21]. The relevant advantage of the NMR approach compared to chromatography–mass spectrometry was the unique properties of NMR of having the same response factor for different classes of metabolites. As a first step, we obtained the ^1^H NMR spectra of the three extracts and we observed a different behavior of the dormant sample compared with vegetative and flowering stage samples. Only in the latter two the signals in the aliphatic region (0.5–2.5 ppm) and sp^2^ region (5–8 ppm), were significant as shown in Figure 1.

Superimposition of the ^1^H-NMR spectra revealed some differences, particularly in the extracts from the vegetative and flowering stages compared to the dormant stage, which revealed a high number of peaks. Thus, investigations to assess the preliminary composition of the extracts were performed using the flowering stage sample to acquire 1D and 2D NMR measurements. Principal signals are listed in Table 1, showing the ^1^H, ^13^C resonances and principal homonuclear (obtained by COSY spectrum) and heteronuclear correlations (obtained by combination of HMBC and HSQC-DEPT measurements) as well as tentative assignments to the different classes of phytoconstituents on the basis of previously published compound assignments [22,23,24,25]. Signals related to aromatic and phenolic constituents can be recognized, those ascribable to hydroxycinnamic acid derivatives, and signal correlating with carbonyl function (δ 166.8–165.3) that is ascribable to ester linkages due to the presence of the trans double bond. Furthermore, signals supporting the presence of flavonol O-glycosides can be observed, namely the proton of the o-p disubstituted aromatic ring and several signals ascribable to H-6 and H-8 as well as H-3 of the chromanone nucleus. Carbohydrate and glycoside substituents can be observed due to the partially overlapped doublets in the range δ 4.7–5.10 showing HSQC-DEPT correlations with CH carbons in the range of δ 90–105 that support the presence of anomeric position of sugars. Several proton signals are observed in the spectrum region of δ 3.00–4.00, showing correlations with CH and presenting single couplings in the COSY spectrum with proton signals at δ 3.30–3.60 ppm and showing TOCSY correlations, suggesting the existence of five proton signals coupling together. Thus, these spin systems are related to the presence of hexose units. Comparison of COSY, TOCSY and HSQC-DEPT data supports the presence of glycoside units. The presence of glycosydic linkage was observed thanks to shift of carbon resonances at δ 80–81 as well as due to HMBC correlations observed from the proton signals with anomeric carbon resonances. Also, CH_2_ groups ascribable to position six of hexose (as glucose) are detected both in free and glycosidic form that are distinguished based on their carbon resonance (Table 1). Furthermore, signals supporting the presence of terpenoids are detected, in particular the singlets supporting the presence of methyl groups correlating with carboxyl function. Other significant resonances are sp^2^ CH_2_ and other relevant signals in the aliphatic part (Figure 2).

Thus, NMR experiments indicated the presence of phenolics, especially caffeic acid derivatives, as well as flavonol glycosides, both C and O derivatives. Based on NMR data, vegetative and flowering stage extracts differ from the dormant stage extract in the presence of terpene derivatives.

Starting from the preliminary NMR data a specific LC-DAD-MS^n^ method in negative ion mode was performed to have a detailed identification of constituents. LC-DAD-MS^n^ analysis of the three extracts confirmed the presence of different classes of polyphenols namely hydroxycinnamic [26] and phenylethanoid derivatives [27], O-glycosylated and C-glycosylated flavonoids [28,29], and gallic acid derivatives that were identified by comparing their UV-VIS and MS^n^ spectra with available standards or literature data. Some phenylethanoid glycosides, which are characteristic compounds in other Lamiaceae, were identified, and the most abundant ones were echinacoside, verbascoside and isoverbascoside [23] (Table 2). Furthermore, O-glycosylated and C-glycosylated flavonoids, mostly derivatives of luteolin, apigenin and kaempferol, were detected.

The samples at vegetative and flowering stages presented similar qualitative composition (Figure 1). Indeed, in these extracts the number and amount of phytoconstituents was higher than that of dormant stage (Table 2). In detail, total flavonoids were 0.006, 0.703 and 1.053% in the extracts obtained from the dormant, vegetative, and flowering stages, respectively. Some O-glycosylated and C-glycosylated flavonoids, mostly derivatives of apigenin and luteolin, were not detected in the dormant stage extract. Variation is observed for the compounds that are present in larger amounts in the extract, namely the labdane derivatives that were present at 0.26, 20.5 and 20.6% in the dormant, vegetative, and flowering stages respectively. The most abundant compounds were isoverbascoside, echinacoside and verbascoside.

At retention times longer than 16.1 min four intense peaks lacking the UV-VIS spectrum ascribable to phenolic compounds were detected. Three of them resulted in ion species that in MS^2^ present a fragment at *m/z* 331 in negative ion mode, and one showed [M − H]^−^ at *m/z* 331. Thus, isolation of these compounds was performed using *R. michauxii* extracts at the vegetative and flowering stages. The first isolated compound was characterized by MS spectrum showing [M − H]^−^ at *m/z* 331. Extensive analysis of 1D and 2D NMR spectra showed the presence of two quaternary methyl groups, eight aliphatic CH_2_ and one olefinic sp^2^. Furthermore, two aliphatic CH were observed and one sp^2^ CH. HMBC, COSY and NOESY correlations allowed the identification of its structure as an ent-labdane derivative that, after comparison to the literature, compound **1** was identified as ent-labda-8(17),13-dien-18-oic acid-15,16-olide (Figure 3), as previously reported [30,31]. ^1^H and ^13^C assignments are reported in Appendix A.

The observed MS fragmentation of the compound **1** (331 *m/z*) [M–H]^−^ can be rationalized as proposed in the scheme (Appendix A). Formal loss of CO (−28), CO_2_ (−44) and cleavage of the ethyl bridge were observed (Appendix A).

Compound **2** was isolated as an oily residue; the MS spectrum in negative ion mode presented the [M−H]^−^ ion at *m/z* 493 and the main fragment was formed by the loss of a hexose unit leading to the ion at *m/z* 331 that showed the same fragmentation pattern of compound **1** (Table 2) in MS^3^. The ^1^H-NMR spectrum presented a large superimposition to compound **1** suggesting that compound **2** shared the same labdane skeleton. Diagnostic HMBC correlations were observed from the methyl group 20 (δ 0.79), with carbon resonances at δ 38.0 (C-10), 49.3 (C-5), 37.7 (C-1), 56.5 (C-9) as well as from the methyl group 19 at δ 1.20, with carbonyl function at C-18 (δ 177.4), with quaternary carbon at δ 47.4 (C-4), a CH_2_ at δ 36.2 (C-3) and C-5. COSY correlations are observed from the methylene at δ 1.89–1.21 (H-1) with signals at δ 1.66 (H-2) and from the latter to signal at δ 1.85–1.63 (H-3). All the observed data allowed the assignment of the first six member ring of the compound and supported the presence of an ester group at position C-18. The second ring of the diterpene was deduced by observing the HMBC correlations from the exocyclic sp^2^ methylene at δ 4.92–4.58 (H-17) with C-8 (δ 146.0), C-9 (δ 56.5), and C-7 (δ 37.2). Correlations from the methyl group 20 (δ 0.79) with C-9, C-10, C-5 support the presence of 1,4a-dimethyl-6-methylene-5-(2-(5-oxo-2,5-dihydrofuran-3-yl) ethyl) decahydronaph-thalene-1-carboxylic acid as part of the compound. The side chain was formed by a five-member lactone ring linked with an ethyl bridge to C-9. Diagnostic HMBCs were observed from the proton signal at δ 5.91 (H-14) with C-12 (δ 27.4), C-15 (δ 174.1) and C-16 (δ 73.5), supported the presence of a lactone and a linkage to an ethyl chain. Long range correlations from signal at δ 2.63–2.33 (H-12) both with signals of the five-member ring as well as with C-9 support the linkage of the two parts. Furthermore, the HMBC correlations were observed from the proton signal at δ 1.83 (H-9) with C-11 (δ 21.1), C-12 (δ 27.4) allowed to confirm the linkage of the side chain in C-9. NOESY correlations revealed the same orientation for methyl group 20 and side chain as well as for methyl group 19.

The differences in spectra of compound **2** compared to **1** are related to the presence of one anomeric proton signal visible as a doublet at δ 5.45 (*J =* 8.0) that correlates with a carbon resonance at δ 94.8 in HSQC and presents HMBC correlations with the ester carbonyl group at δ 177.4 (C-18). Thus, on the basis of HSQC, HMBC, COSY and NOESY data, the structure of the sugar residue was ascribed to a *β*-glucopyranosyl unit as indicated in Appendix A, and the structure of compound **2** was assigned to ent-labda-8(17),13-dien-18-glucopyranosyl ester-15,16-olide. Hydrolysis and derivatization of the sugar with R-butanol allowed the establishment of its absolute configuration as D. To our knowledge this is the first report of the glucoside of the ent-labda-8(17),13-dien-18-oic acid-15,16-olide. Recently, Hu et al. [32] reported the isolation of some diterpenoids from *Pinus kwangtungensis* and described the β-D-glucopyranosyl- (4S,5R,9S,10R)-labda-8(17),13-dien-15,16-olid-19-oate. The comparison of the NMR data with our assignments revealed differences in the chemical shifts of the position five as well as of the methyl group linked to C-4 thus indicating different stereochemistry for these positions. We assumed that our compound shared the absolute configuration previously reported by Zdero [30] and by Henrich and Jefferies [31].

Compound **3** was isolated as an oily residue, the MS spectrum in negative ion mode presented the [M + H]^−^ ion at *m/z* 361, and the formic acid adduct [M + FA − H]^−^ ion at *m/z* 407. Main fragments were observed at *m/z* 199 (−162 Da) suggesting the loss of one hexose unit, and the *m/z* 179 (−182) (Appendix A). The compound presents an ^1^H-NMR spectrum supporting the presence of a glycosidic iridoid. The presence of a *β*-glucopyranosyl unit was deduced by the anomeric proton signal at δ 5.43 (*J =* 8.2) δ 94.0 (H-1′). The spin system from H-1′ and the assignment of resonances of each position allowed us to establish by HSQC, HMBC, COSY and NOESY correlations, the presence of a beta-glucopyranosyl unit (See Appendix A). Furthermore, data from HSCQ and HMBC allowed us to observe a further nine carbon signals, one hemiacetalic δ 5.41 (*J* = 8.5) δ 94.6, and one quaternary at δ 1.56 (δc 16.6). Furthermore, there were three aliphatic CH at δH 3.85 δC 77.1 (C-6), δH 3.34 δC 64.0 (C-7), δH 2.27 δC 51.5 (C-9). Long range HMBC correlations allowed the observation of a quaternary carbon at 65.7 (C-8). Diagnostic HMBC correlations were observed from H-4 with C-6, C-9 and from H-9 (δ 2.27) with C-4, C-8, C-1 and from the methyl group C-10 (δ 1.56) with C-8, C-7 and C-9 supporting the presence of cyclopentan-pyran iridoid nucleus. The coupling constant of H-1 (8.5 Hz) and the NMR assignments confirm the structure of antirrhinoside [33]. The quantification of compound **3** was perform by NMR. Antirrhinoside was present in large amounts in the vegetative and flowering stages, 7.7 and 5.5%, respectively, while it was not detectable in the dormant stage.

The semipreparative HPLC also allowed the isolation of isoverbascoside (**4**), echinacoside (**5**) and verbascoside (**6**); their structures were confirmed by comparison of their spectral data with previously published data [27,34]. The isolation of such compounds confirmed their large presence in the extracts as observed in the NMR analysis of crude extracts at the flowering stage (Table 1) and by LC-DAD-MS (Table 2).

### 2.2. Cytotoxic Effect on Hepatocarcinoma Cell Line Huh7 of Compounds Isolated from R. michauxii Extracts

To obtain some information about the potential bioactivity of *R. michauxii*, the effect of the isolated compounds, namely ent-labda-8(17),13-dien-18-oic acid-15,16-olide (**1**), ent-labda-8(17),13-dien-18-glucopyranosyl ester-15,16-olide (**2**), antirrhinoside (**3**), isoverbascoside (**4**) echinacoside (**5**) and verbascoside (**6**), was investigated on the expression of the LDL receptor in the hepatocarcinoma Huh7 cell line. In a first series of experiments, tests of the potential cytotoxicity of the six isolated compounds were performed. As shown in Figure 4, no significant cytotoxic effect was observed at concentration up to 50 µM.

### 2.3. Compounds Isolated from R. michauxii Extract Increases LDLR and PCSK9 Expression in the Huh7 Cell Line

The effects of the compounds isolated from *R. michauxii* extract on the expression of LDLR in Huh7 cells were tested. Cells were incubated with two concentrations of the extracts (25 and 50 µM) for 72 h in MEM/10% FCS using simvastatin 2.5 µM as a positive control. As shown in Figure 5, a significant increase in LDLR expression was observed after incubation with **3** at both concentrations and with 2 at 50 µM. Interestingly, the induction observed after the incubation with 50 µM of **3** was similar to that observed with simvastatin (3.7 ± 2.2 fold vs. control and 5.9 ± 3.2 fold vs. control for **3** and simvastatin, respectively).

Since compounds (**2**) and (**3**) showed a significant induction of the LDLR, we explored their potential action on one of the main cholesterol homeostasis regulators, proprotein convertase subtilisin/kexin type 9 (PCSK9). Similar to simvastatin, compound **3** significantly induced PCSK9 by 3.2 ± 1.5-fold vs. control (Figure 6).

These data suggest that (**3**) may inhibit, similarly to simvastatin, cholesterol biosynthesis and thus, in turn, activate the transcription of sterol regulatory element binding protein (SREBP)-transcriptionally regulated genes, such as the LDL receptor and PCSK9. To further explore this possibility, we determined, by LC-MS, the intracellular concentration of cholesterol after 72 h of treatment with simvastatin (control) and compounds (**1**), (**2**) and (**3**). As expected, 0.5 µM simvastatin resulted in a significant reduction of intracellular cholesterol (−46.3 ± 2.2% vs. control). A similar effect was observed in response to compound (**3**) that resulted in a significant reduction of intracellular cholesterol (−38.3 ± 2.2% vs. control) (Figure 7). Taken together, it is tempting to speculate that compound (**3**) may affect cholesterol biosynthesis in Huh7 cells; an effect that resulted in the increase of the LDLR and PCSK9 levels.

## 3. Discussion

The results of the present work revealed that *R. michauxii* (Briq.) Scheen and V.A.Albert can be a valuable source of different classes of phytoconstituents as shown by LC-DAD-MS^n^ and NMR data. Due to the higher amount of phytoconstituents, the vegetative and flowering stages of the plants were used for isolation of phytoconstituents. The main compounds isolated were phenyletanoids, labdanes and iridoids. Literature has reported the isolation of a series of labdanes in different *Ridingia* species [5,6,35] and our work highlights the presence of other classes of compounds in *R. michauxii,* demonstrating the synthetic ability of this species to produce various derivatives. Considering the high relevance of diterpenes in this plant, only minor information is present in the literature about phenolic composition [5,6,35]. Our chemical characterization allowed the identification of eleven hydroxycinnamic derivatives with isoverbascoside, echinacoside and verbascoside being the most abundant phenolics in the vegetive and flowering stage. The flavonoid fingerprint was made up of five C-glycosylated flavonoids and eight O-glycosylated flavonoids and all of them were apigenin or luteolin derivatives. The total amount of flavonoids were ten times less abundant compared to hydroxycinnamic derivatives. Compounds in these extracts were chemically heterogeneous with the presence of phenolics, the lignan medioresinol, four labdane derivatives, one iridoid, and the antirrhinoside. Based on the traditional uses of this plant for different ailments, metabolic diseases and cardiovascular problems we have explored the effects of the main constituents on the LDL and PCSK9 targets.

*Otostegia persica* extracts (*R. persica*) have been studied using in vivo models and presented similar effects to atorvastatin in decreasing serum lipids but not high-density lipoprotein (HDL), oxidative stress factors, aortic contraction, weight gain, or blood pressure [36]. Unfortunately, in that study information about the composition of the extracts were missing. The results obtained from our in vitro model on LDL-R and PCSK9 showed that ent-labda-8(17),13-dien-18-oic acid-15,16-olide (**1**), ent-labda-8(17),13-dien-18-glucopyranosyl ester-15,16-olide (**2**), echinacoside (**4**), isoverbascoside (**5**) and verbascoside (**6**) have no effect or limited effects on these targets at the tested doses. On the other hand, antirrhinoside (**3**), at tested doses, was able to interact with LDL-R and PCKS9 indicating that this compound can modulate these specific proteins that are involved in cholesterol balance. Furthermore, the compound (**3**) when incubated with hepatocarcinoma cells (at 25 and 50 µM) reduced the total cholesterol in a comparable amount to simvastatin at 0.5 µM.

Iridoids have been described as active compounds possessing significant cardiovascular, hypoglycemic, hypolipidemic, antihepatotoxic, choleretic, anti-inflammatory, antispasmodic, bioactivities [37], but their role in specific pathologies still needs to be investigated.

Considering the literature on other iridoid compounds, geniposides have been reported as potential active compounds in cholesterol metabolism and atherosclerosis. In ApoE-knockout mice oral administration of geniposide lowered both plasma lipid levels and dendritic cell numbers [38]. Furthermore, in another study geniposide was able to protect against atherosclerosis and inhibited the formation of foam cells by regulating the equilibrium of expression of diverse lipid transporters in ApoE-knockout mice [39]. Moreover, geniposide accelerated the hepatic synthesis of bile acids that inactivate the FRX mediated negative feedback regulation of bile acids [40]. This induces increases in reverse cholesterol transport and cholesterol catabolism. In the same paper the authors reported that geniposide reduced ileal FXR-mediated reabsorption of bile acids, resulting in increased excretion of bile acids [40]. Gentiopicroside, in both acute and chronic alcohol induced mouse hepatosteatosis (the later stage of alcoholic liver disease, ALD) models, mitigated the upregulation of SREBP-1 and downregulation of PPAR-α, among others, by activation of AMPK [37,41].

Our results indicate that antirrhinoside presents significant activity as a LDL-R inducer and is able to reduce intracellular cholesterol levels in hepatic cells, showing new potential mechanisms of action of iridoids as cholesterol lowering agents.

Thus, several lines of evidence suggest a positive effect of iridoid compounds on lipid metabolism and atherosclerotic plaque development and our in vitro data identified an additional compound of the same chemical class with a potential hypocholesterolemic effect by a statin-like mechanism, probably related to the inhibition of the mevalonate pathway.

Although additional in vitro and in vivo analyses should be performed to define the molecular mechanism of action of antirrhinoside (**3**) and its actual hypocholesterolemic effect, it is tempting to speculate that this compound may interfere with some enzymatic reaction in the mevalonate pathway. This inhibition could be responsible for the induction of both the LDL receptor and the PCSK9, as observed with the HMG-CoA reductase inhibitor simvastatin. Additional studies will be required to define its pharmacological action in an in vivo model of atherosclerosis.

## 4. Materials and Methods

### 4.1. Plant Materials

*R. michauxii* was collected at three phenological stages in Kazeroun, Fars province, Iran, i.e., on 5 February (dormant stage), 13 March (vegetative stage) and 12 April (flowering stage), 2018. The plant samples were confirmed by Prof. Ahmad Reza Khosravi with the voucher specimen (NO. 55083) and deposited at the Herbarium of Shiraz University.

### 4.2. Extract Preparation

Plant parts were air-dried at room temperature in the dark conditions and ground with a blender into fine powder. The plant powder was soaked in 80% methanol at a ratio of 1:10 (ground plant material to methanol) and kept on a rotary shaker for 48 h at 22 °C. The crude methanolic extract was then filtered using Whatman paper and then dried in a rotary evaporator at 40 °C. Dried extracts were stored at 4 °C until further studies.

### 4.3. Phytochemical Characterization, NMR, HPLC-DAD-MS^n^, Optical Rotation Power Measurament

For the NMR analysis, 30 mg of *R. michauxii* extracts were dissolved in 1 mL of deuterated methanol and sonicated for 5 min. Different spectra were acquired using ^1^H NMR, HSQC-DEPT, HMBC, COSY and TOCSY experiments. NMR spectra were obtained on a Bruker Avance III 400 Ultrashield spectrometer with superconducting 400 MHz magnet at a temperature of 25 °C. They were acquired in MeOD-d4 (Sigma-Aldrich, St. Louis, MO, USA) using Duran^®^ 4.95 mm NMR tubes (Duran Group). Chemical shifts were expressed in δ values in ppm. ^1^H-NMR and HSQC-DEPT, HMBC, COSY experiments were acquired using standard Bruker sequences measuring p1 and d1 for each acquired sample. For the measurement of optical rotation power of isolated compounds, a Jasco digital polarimeter P2000, was used.

For the phytochemical characterization of the *R. michauxii* extracts, 50 mg of each extract was dissolved in 20 mL of methanol and sonicated for 10 min in an ultrasonic bath at room temperature. The solution was centrifuged for 5 min and the supernatant was used for HPLC-DAD-MS analysis. Polyphenols were quantified using Diode Array Detector (DAD) data classifying the different constituents based on their UV/Vis spectra in flavonoids, hydroxycinnamic acids and gallic acid derivatives. Qualitative–quantitative analysis was obtained by HPLC-DAD-MS^n^. The measurements were performed with an Agilent 1260 chromatograph (Santa Clara, CA, USA) equipped with a 1260 diode array (DAD) and Varian MS-500 ion trap (Santa Clara, CA, USA) as detectors. Separation was achieved using a Synergy Polar-RP (150 × 3 mm, 4 µm) (Phenomenex, Torrance, CA, USA) as stationary phase. The mobile phases were water, 0.1% formic acid (A) and acetonitrile (B). The elution gradient started at A 90%, then A% decreased to 0% over 30 min. The flow rate was 0.4 mL/min. At the end of the column a T connector split the flow rate to DAD and MS. The DAD detector was used to quantify flavonoids and polyphenols. Namely, rutin, vitexin, chlorogenic acid, and gallic acid were used as reference compounds for O-glycosylated flavonoids, C-glycosylated flavonoids, hydroxycinnamic acid and gallic acid derivatives, respectively. UV-Vis spectra were acquired in the range of 200–650 nm and chromatograms were monitored at 350, 330 and 280 nm for the quantification of flavonoids, hydroxycinnamic acid and gallic acid derivatives, respectively. The sample injection volume was 10 µL. MS spectra were recorded in negative ion mode in the range of *m*/*z* 50–2000, using an Electrospray (ESI) ion source. Fragmentation of the main ionic species were obtained by the turbo data dependent scanning (TDDS) function. The identifications of compounds were obtained on the basis of fragmentation spectra as well as comparison of fragmentation patterns with those reported in the literature. In addition, the comparison with reference compounds available in the authors’ laboratory was used whenever possible. Quantification of phenolic constituents was obtained by the calibration curve method. The equations obtained for the various groups of phenolic compounds were as follows: rutin y = 29.058x + 78.235 (R^2^ = 0.998); vitexin y = 105.3x + 50.231 (R^2^ = 0.997); chlorogenic acid y = 53.787x + 7.742 (R^2^ = 0.999); gallic acid y = 34.215x + 25.473 (R^2^ = 0.999).

### 4.4. Isolation and Structural Elucidation of Phytoconstituents: Semipreparative HPLC, Determination of the Absolute Configuration of the Sugar Residues after Hydrolysis

For the purification of phytoconstituents, 3 g of extracts obtained from the vegetative and flowering stages were pooled and dissolved in methanol. Separation was performed in a Sephadex column (4 × 70 cm) and methanol was used as mobile phase at 0.5 mL/min. From the column, 20 fractions were collected and pooled in 5 fractions on the basis of TLC behavior. More hydrophilic fractions pooled on the basis of TLC behavior were purified by semipreparative HPLC on an Eclipse C18 21.2 × 150 mm, 5 μm (Agilent Technologies, Santa Clara, CA, USA) semipreparative column using an Agilent 1260 chromatograph (Santa Clara, CA, USA) equipped with 1260 diode array (DAD). Mobile phases were methanol and water 1% formic acid (30/70) with isocratic gradient, at flow rate of 2.5 mL/min. Echinacoside (7.7 mg), isoverbascoside (5.5 mg) and verbascoside (6.1 mg) were isolated and their structure confirmed by NMR comparisons with reference compounds. Fractions 2 and 3 were pooled into a single fraction due to their similar chemical profile. This fraction was eluted in a silica column (5 × 30 cm) using hexane and a mixture of hexane/ethyl acetate as the eluting system. From fractions 19-25, the ent-labda-8(17),13-dien-18-oic acid-15,16-olide (25 mg) was isolated, and the ent-labda-8(17),13-dien-18-oic acid-15,16-olide glucopyranoside (14.5 mg) from fractions 54–56. The antirrhinoside (13.5 mg) was isolated from fractions 118–125. The structures of the isolated compounds were confirmed by 1D and 2D NMR spectroscopy and MS/MS experiments. The amount of labdane derivatives was determined using the isolated ent-labda-8(17),13-dien-18-oic acid-15,16-olide as a reference standard using the LC-DAD-MS method described in Section 4.3 and quantifying the compound with 331 *m/z* signal in negative ion mode. The calibration curve was y = 1688.6x − 6593.3 (R^2^ = 0.989). The quantitative measurement of antirrhinoside was obtained by NMR using the H-3 signal of the iridoid for quantitative purposes, compared to the internal standard using a method previously described [42,43]. In detail, caffeine was used as an internal standard for quantitative ^1^H-NMR measurements. A stock solution of caffeine in deuterated chloroform was prepared and 1000 μL of this solution (0.5 mg/mL) was added to an Eppendorf tube with 10 mg of extract powder exactly weighed. The mixture was sonicated for 10 min or longer to complete solubilization, then centrifuged and liquid transferred to the NMR tube for measurement.

Compound **1**: ent-labda-8(17),13-dien-18-oic acid-15,16-olide (25 mg), NMR data see Appendix A. [α]_D_^20^ + 7.4 (*c* 0.1, CHCl_3_).

Compound **2**: ent-labda-8(17),13-dien-18-oic acid-15,16-olide glucopyranoside (14.5 mg). NMR see Appendix A. [α]_D_^20^ + 4.2 (*c* 0.14, MeOH).

Compound **3**: antirrhinoside (13.5 mg). NMR see Appendix A. [α]_D_^20^ -68 (*c* 0.10, MeOH).

For the determination of the absolute configuration of the monosaccharides, a sample (1 mg) of the compounds was treated with (R)-2-butanol (0.5 mL and AcCl (0.1 mL)) at room temperature for 24 h. The solvent was evaporated, then the residue was acetylated using acetic anhydride in pyridine overnight. The produced (R)-2-butanol ester acetylated derivatives were compared with those of standard samples prepared from the respective monosaccharides [44].

### 4.5. In Vitro Experiments

#### 4.5.1. Reagents

Eagle’s minimum essential medium (MEM), trypsin-EDTA, penicillin, streptomycin, sodium pyruvate, L-glutamine, nonessential amino acid solution, fetal bovine serum (FBS), plates, and Petri dishes were purchased from EuroClone (Milan, Italy). *R. michauxii* isolated compounds were diluted or dissolved in dimethyl sulfoxide (DMSO, Sigma-Aldrich) as a stock solution of 0.02 µM. Simvastatin (Merck, Sharp, and Dohme Research Laboratories, Kenilworth, NJ, USA) was dissolved to a stock concentration of 50 mM in 0.1 M NaOH, and the pH was adjusted to 7.2 according to manufacturer’s instructions. The solution was then sterilized by filtration.

#### 4.5.2. Cell Cultures

Human hepatic cancer cells (Huh7) were cultured in MEM supplemented with 10% Fetal Bovine Serum (FBS), 1% L-glutamine 200 mM, 1% sodium pyruvate 100×, 1% nonessential amino acids 100×, and 1% penicillin/streptomycin solution (10,000 U/mL and 10 mg/mL, respectively), at 37 °C in a humidified atmosphere of 5% CO_2_ and 95% air. For the experiments, cells were incubated with the indicated final concentrations in MEM/10% FBS. The final concentration of solvent (DMSO) did not exceed 0.5% *v*/*v* and the same amount was added to all of the experimental points in each assay.

#### 4.5.3. Cell Viability Assay on Huh7

Cells were seeded in MEM/10% FBS in a 96-well tray, at a cellular density of 8000 cells/well. The day after, treatments were added (four experimental points for each com-pound; 50 µM, 25 µM, 12.5 µM and 6.25 µM) for 72 h, after which the cell viability was evaluated by the sulforhodamine B assay (SRB) according to a previously published protocol.

#### 4.5.4. Western Blot Analysis

Huh7 cells were seeded in MEM/10% FBS in 6-well trays, at the cellular density of 300,000 cells/well. The day after, the medium was replaced with the compounds at the indicated concentrations in DMEM/10% FBS. After 72 h of incubation, intracellular protein content was extracted in lysis buffer (50 mM Tris pH 7.5, 150 mM NaCl, and 1% Nonidet-P40, containing 1% *v*/*v* of protease and phosphatase inhibitor cocktails). Protein samples (25 µg) and a molecular mass marker (Thermo Scientific, Waltham, MA USA) were separated using 4–20% SDS-PAGE (Bio-Rad, Hercules, CA, USA) under denaturing and reducing conditions. The protein samples were then transferred to a nitrocellulose membrane using the Trans-Blot^®^ Turbo™ Transfer System (Bio-Rad) and nonspecific binding sites were blocked with a 5% non-fat dried milk tris-buffered tween 20 (TBS-T20) solution, with agitation for 60 min at room temperature. The blots were incubated overnight at 4 °C with a diluted solution (5% nonfat dried milk) of anti-LDLR (rabbit polyclonal antibody, GeneTex GTX132860; dilution 1:1000), anti-PCSK9 (rabbit polyclonal antibody, GeneTex GTX129859; dilution 1:1000), anti-GAPDH (rabbit polyclonal antibody, GeneTex GTX100118; dilution 1:3000). The membranes were washed with TBS-T20 and exposed for 90 min at room temperature to a di-luted solution (5% nonfat dried milk) of the secondary antibodies (peroxidase-conjugate goat anti-rabbit and anti-mouse, Jackson ImmunoResearch, dilution 1:5000, cod. 111-036-045 and 115-036-062, respectively). Immunoreactive bands were detected by exposing the membranes to ClarityTM Western Enhanced ChemiLuminescence (ECL) chemiluminescent substrates (Bio-Rad) for 5 min, and images were acquired with an Azure c400 Imaging System (Aurogene, Rome, Italy). The densitometric readings were evaluated using ImageLabTM software (Bio-Rad).

#### 4.5.5. Cholesterol Determination

Huh7 cells were incubated under the same experimental conditions described for western blot analysis. At the end of the period of incubation, cell monolayers were washed with PBS (phosphate buffer saline) and incubated for 2 h at RT with 0.1 N NaOH. An aliquot of these samples was used for protein determination (BCA assay) and a second aliquot for cholesterol analysis.

The total cholesterol content of cells treated with the different compounds was measured using liquid chromatography coupled with mass spectrometry with atmospheric pressure chemical ionization ion source (LC-APCI-MS). The system for analysis was an Agilent 1260 Liquid chromatograph, coupled with a Varian mass spectrometer MS 500 with ion trap analyzer. As the ion source, an atmospheric pressure chemical ionization (APCI) was used operating in positive ion mode. Spectra were acquired in the range *m/z* 350–550. Cholesterol was detected as [M-H_2_O]^+^ species at *m/z* 369.5. A calibration curve was created in the range of 120.0–0.5 ug/mL with solutions at four concentration levels. Samples were prepared as follows: cells were treated with NaOH (0.1 M) for 1 h at 60 °C before injection. Samples were then diluted with equal volume of DMSO and used for chromatography.

#### 4.5.6. Statistical Analysis

Data are expressed as mean ± standard deviation. Differences between the two groups were analyzed via *t*-test analysis (GraphPad, San Diego, CA, USA). *p*-values lower that 0.05 were considered statistically significant.

## 5. Conclusions

A comprehensive phytochemical characterization of *R. michauxii* (Briq.) Scheen and V.A.Albert extracts by NMR and LC-DAD-MS^n^ is reported here, and shows the presence of various types of polyphenols, including phenylethanoid glycosides and O-glycosylated and C-glycosylated flavonoids, labdane diterpenes and iridoids. The most abundant compounds in the vegetative and flowering stages were the labdane diterpenes accounting for 20% of the total extract. Main compounds present in the extract were isolated and tested on the LDLR and PCSK9 targets. In vitro studies on the human hepatocarcinoma cell line, Huh7, predicts a possible hypocholesterolemic effect of compound **3** with a statin-like mechanism, possibly by inhibiting the mevalonate pathway. Thus, antirrhinoside (**3**) may represent a new naturally occurring compound with potential hypocholesterolemic action.

## Figures and Tables

**Figure 1 molecules-27-02256-f001:**
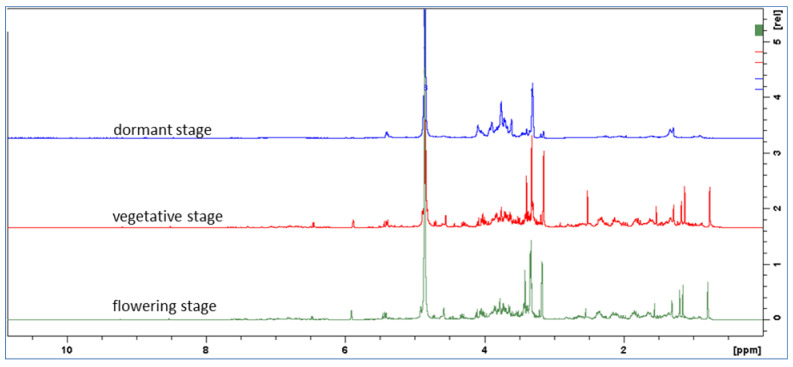
^1^H NMR of extracts at dormant, vegetative and flowering stages (blue, red and green) in MeOD.

**Figure 2 molecules-27-02256-f002:**
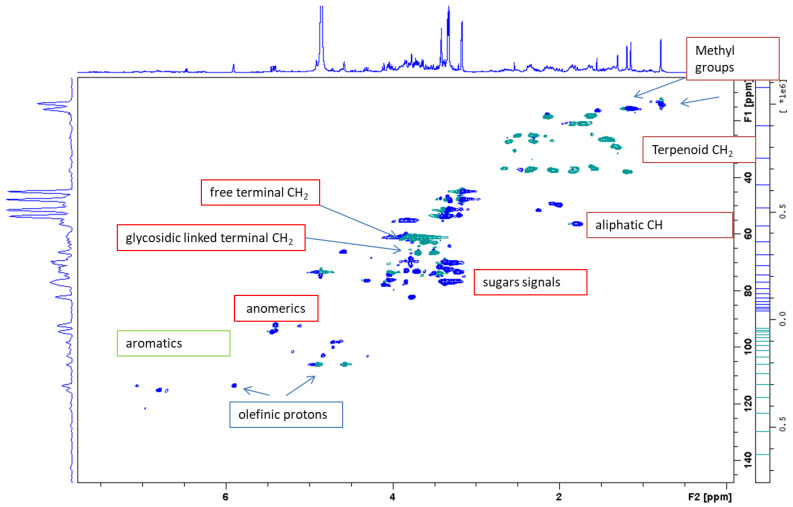
Enlargement of the HSQC-DEPT spectra of *R. michauxii* extracts at the flowering stage in MeOD_4_.

**Figure 3 molecules-27-02256-f003:**
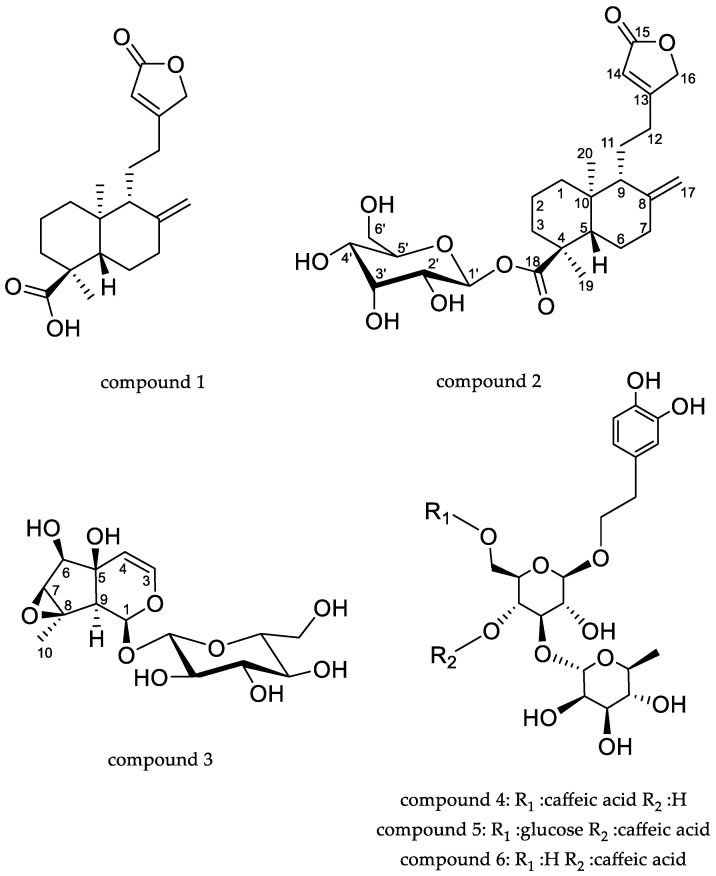
Chemical structures of compounds isolated from *R. michauxii* extract. Compound **1**: ent-labda-8(17),13-dien-18-oic acid-15,16-olide; compound **2**: ent-labda-8(17),13-dien-18-glucopyranosyl ester-15,16-olide; compound **3**: antirrhinoside, compound **4**: isoverbascoside; compound **5**: echinacoside; compound **6**: verbascoside.

**Figure 4 molecules-27-02256-f004:**
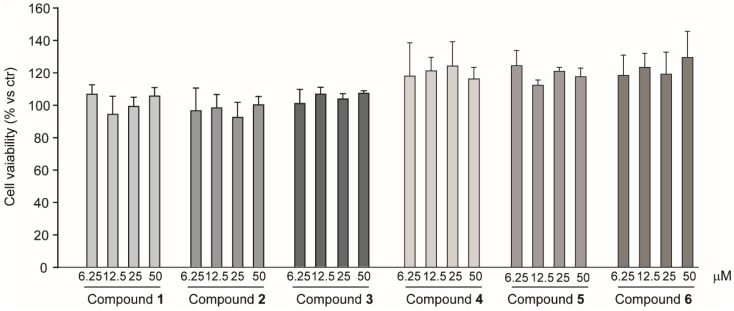
Cytotoxicity effect of compounds isolated from *R. michauxii* in Huh7 cell line. Cells were seeded (8000/well of 96 well tray) and incubated with DMEM supplemented with 10% FCS; 24 h later the medium was replaced with one containing 10% FCS and the reported concentrations of compounds and the incubation was continued for an additional 72 h. At the end of this incubation period the cell viability was determined by the SRB assay. Each bar represents the mean ± SD of three independent experiments.

**Figure 5 molecules-27-02256-f005:**
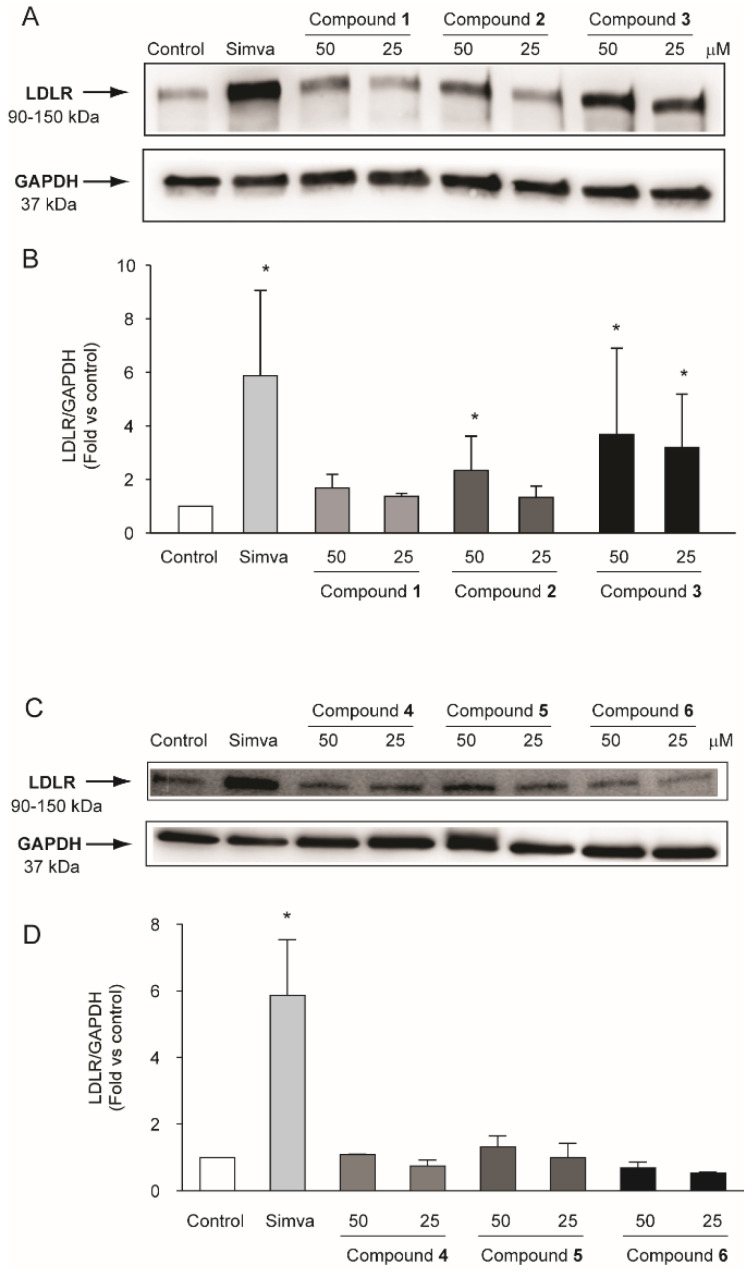
Effect of compounds isolated from *R. michauxii* on LDLR expression in the Huh7 cell line. Cells were incubated with MEM/10% FCS in the presence or absence of indicated concentrations of compounds and 2.5 µM simvastatin (simva). After 72 h, total protein extracts were prepared and LDLR expression evaluated by western blot analysis. GAPDH was used as a loading control. (**A**,**C**) representative images of three independent experiments. (**B**,**D**) densitometric readings were evaluated using ImageLab^TM^ software. Each bar represents the mean ± SD of three independent experiments. * *p* < 0.05 vs. control.

**Figure 6 molecules-27-02256-f006:**
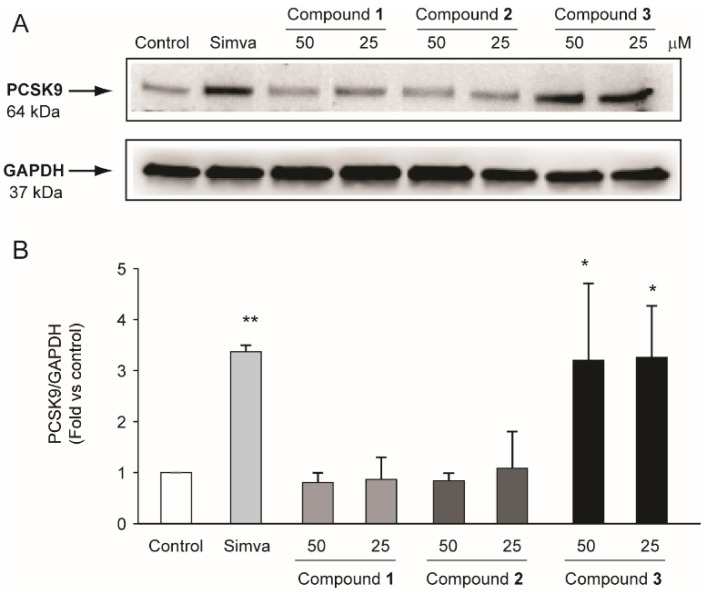
Effect of compounds **1**–**3** isolated from *R. michauxii* on PCSK9 expression in the Huh7 cell line. Cells were incubated with MEM/10% FCS in the presence or absence of indicated concentrations of compounds and 2.5 µM simvastatin (simva). After 72 h, total protein extracts were prepared and PCSK9 expression evaluated by western blot analysis. GAPDH was used as a loading control. (**A**) representative images of three independent experiments. (**B**) densitometric readings were evaluated using ImageLab^TM^ software. Each bar represents the mean ± SD of three independent experiments. * *p* < 0.05; ** *p* < 0.01 vs. control.

**Figure 7 molecules-27-02256-f007:**
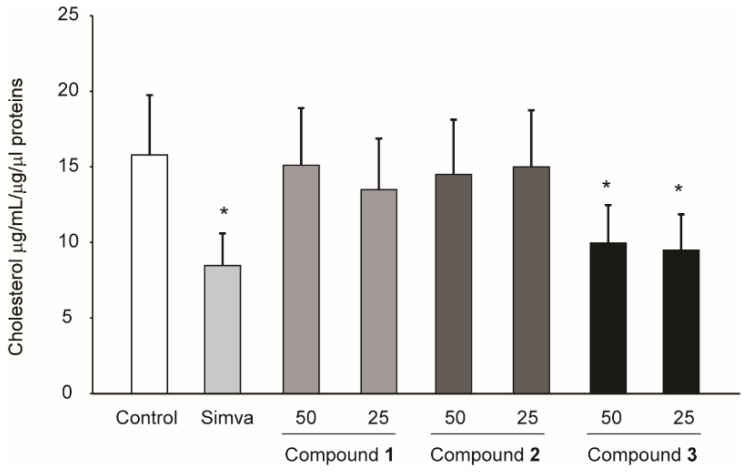
Effect of compounds **1**–**3** isolated from *R. michauxii* on intracellular cholesterol of the Huh7 cell line. Cells were incubated with MEM/10% FCS in the presence or absence of indicated concentrations of compounds and 0.5 µM simvastatin (simva). After 72 h, total lipids were extracted, and cholesterol level determined by mass spectrometry analysis. These values were normalized by protein levels. Each bar represents the mean ± SD of three independent experiments. * *p* < 0.05 vs. control.

**Table 1 molecules-27-02256-t001:** 1D and 2D NMR correlation of the main classes of constituents of *R. michauxii* extract at the flowering stage. * partially overlapped signals.

Class of Constituents and Assignments	δ H	δ C	Significant Correlations
Flavonoids: apigenin and other derivatives			
H 2′-6′ (apigenin)	7.60 m	130.3	HMBC: 162.0 COSY: 6.42
H 3′-5′ (apigenin)	6.42 d (*J =* 8.0 Hz)	115.0	
H-3 (apigenin or other flavonoids)	6.68 s	102.8	
H 6-8 of O-glycosidic flavonoids	6.81 * m	90.7	HMBC: 163.5
	6.78 m	94.7	
	6.59 * m	97.3	
	6.57 m	89.9	
Anomeric signals of sugar residues (O-glycosides)	5.11 d (*J =* 7.5 Hz)	92.7	COSY 3.30–3.41–3.85–4.10
	5.18 * m	101.7	
	5.38 * m	92.3	
	5.42 * m	94.6	
	4.73 * m	98.5	
	4.71 * m	100.0	
	4.67 * m	98.0	
Anomeric signals of sugar residues (C-glycosides)	4.91 * m	73.2	COSY 3.32–3.30–3.46
	4.85 * m	74.1	
Sugar positions of interglycosydic bonds	3.77 m	82.0	COSY 3.30–3.41–3.85–4.10;
	3.51 m	81.8	HMBC: 101.7–98.5
Free CH_2_OH of sugar	3.80–3.53 m	60.5–61.1	
CH_2_OH of sugar with glycosydic linkage	3.71–3.52 m	67.2	
Caffeic acid derivatives, Verbascoside/Echinacoside and similar			
H-7 (double bond)	7.55 d (*J =* 16 Hz)	144.6	HMBC: 165.6- 166.8
	7.48 d (*J =* 16 Hz)	146.9	COSY: 6.30
H-8 (double bond)	6.30 d (*J =* 16 Hz)	113.5	COSY: 7.55–7.48
H-2 (aromatic ring)	7.07 (*J =* 2 Hz)	114.7	
H-6	6.93 m	121.8	
H-5	6.90 m	115.3	HMBC: 147.9
Terpenoids			
Quaternary methyl group	0.75 s	14.3	HMBC: 177.3 49.0 36.3
Quaternary methyl group	1.15 s	16.3	HMBC: 182.4 49.0 36.3
Quaternary methyl group	1.18 s	16.3	HMBC: 56.2 49.1 40.0 36.3
aliphatic CH_2_	1.17 m	38.5	
	1.66 m	18.5	
	2.10–2.45 m	37.6	
	1.68–1.80 m	21.8	
Olefinic proton signals	4.48–4.92	106.5	HMBC: 37.6 146.0 56
	5.90 brs	114.0	HMBC 176.0 21.8

**Table 2 molecules-27-02256-t002:** Qualitative–quantitative composition of hydroxycinnamic derivatives, O-glycosylated flavonoids, C-glycosylated flavonoids and gallic acid derivatives in *R. michauxii* extracts at the dormant, vegetative and flowering stages. Results are expressed as mean ± standard deviation (n = 4). RT: retention time, nd: not detected, ^§^: quantified by NMR. * identification confirmed by reference standard injection, ^#^: *m/z* adduct with formic acid [M + FA − H]^−^. Bold numbers: compound selected for in vitro assay.

RT	[M − H]^−^	Fragmentation	Compound	Dormant Stage %	Vegetative Stage %	Flowering Stage %
			hydroxycinnamic derivative			
9.0	353	191 171	chlorogenic acid *	nd	0.045 ± 0.005	0.166 ± 0.008
10.3	917	755 593 461 315	lavandulofolioside hexoside/samioside hexoside	0.081 ± 0.010	0.079 ± 0.010	0.057 ± 0.002
10.4	785	623 461 315	Echinacoside * (**5**)	0.497 ± 0.018	1.766 ± 0.199	1.633 ± 0.174
11.2	799	623 461 315	feruloyl verbascoside	0.462 ± 0.015	0.546 ± 0.060	0.220 ± 0.046
11.4	931	755	feruloyl samioside	0.019 ± 0.0015	0.031 ± 0.002	nd
11.6	623	461 315	Verbascoside * (**6**)	0.066 ± 0.005	2.364 ± 0.067	4.004 ± 0.122
11.8	755	593 461 315	lavandulofolioside/samioside	0.104 ± 0.010	0.556 ± 0.010	0.758 ± 0.010
12.0	623	461 315	Isoverbascoside (**4**)	0.110 ± 0.010	2.363 ± 0.205	1.145 ± 0.241
12.1	945	769 637 491	unknown phenylethanoid glycosides	0.048 ± 0.009	0.301 ± 0.008	0.154 ± 0.080
12.2	813	637 491	unknown phenylethanoid glycosides	0.057 ± 0.006	0.585 ± 0.080	0.298 ± 0.030
14.5	651	505 475 328	martynoside	0.088 ± 0.0013	0.283 ± 0.020	0.219 ± 0.005
			total amount of hydroxycinnamic derivative	1.531	8.918	8.655
			C glycosylated flavonoid			
9.4	623	533 503 413 383	4′-methoxyluteolin -6,8-di-C-glucopyranoside	nd	0.005 ± 0.001	0.007 ± 0.001
9.6	609	519 489 399 369	luteolin-6,8-di-C-glucoside	nd	0.011 ± 0.002	0.014 ± 0.002
10.2	593	503 473 383 353	apigenin 6,8-di-C-glucoside *	0.004 ± 0.001	0.131 ± 0.010	0.220 ± 0.004
10.8	563	503 473 443 353	apigenin-6-C-glucoside-8-C-xyloside	nd	0.119 ± 0.034	0.149 ± 0.008
11.8	577	487 457 367 337	deoxy apigenin 6,8-di-C-glucoside	nd	0.028 ± 0.006	0.030 ± 0.005
			total amount of C glycosylated flavonoid	0.004	0.295	
			O glycosylated flavonoid			
8.4	417	285 179 163 152	kaempferol pentoside	nd	0.026 ± 0.001	0.049 ± 0.001
9.1	401	269	apigenin pentoside	0.001 ± 0.001	0.004 ± 0.001	0.007 ± 0.001
13.0	447	285 243 199 175	luteolin hexoside	0.001 ± 0.001	0.153 ± 0.001	0.169 ± 0.001
13.4	489	447 285	kaempferol/luteolin acetyl hexoside	nd	0.069 ± 0.001	0.036 ± 0.001
13.4	577	269	apigenin rutinoside (isomer 2)	nd	0.019 ± 0.001	0.030 ± 0.001
13.5	431	269	apigenin-7-O-glucoside *	nd	0.019 ± 0.001	0.034 ± 0.001
14.5	285	241 199 175 154 133	Luteolin *	nd	0.020 ± 0.001	0.026 ± 0.001
15.6	577	269	apigenin rutinoside (isomer 1)	nd	0.099 ± 0.001	0.284 ± 0.001
			total amount of O glycosylated flavonoid	0.002	0.408	0.634
			gallic acid derivative			
10.1	387	207 163	medioresinol	0.026 ± 0.003	2.033 ± 0.087	0.105 ± 0.017
			labdane diterpenoids			
16.1	539 ^#^ 493	331 303 233 221	ent-labda-8(17),13-dien-18-oic acid-15,16- glucopyranoside (**2**)	0.165 ± 0.006	9.387 ± 0.001	10.081 ± 0.090
16.1	825	667 331 303 233 221	ent-labda-8(17),13-dien-18-oic acid-15,16-olide derivative	0.002 ± 0.001	3.662 ± 0.001	2.954 ± 0.090
16.5	529	331 303 233 221	ent-labda-8(17),13-dien-18-oic acid-15,16-olide derivative	0.040 ± 0.005	1.305 ± 0.001	1.089 ± 0.058
17.5	331	303 233 221	ent-labda-8(17),13-dien-18-oic acid-15,16-olide (**1**)	0.057 ± 0.002	6.161 ± 0.302	6.448 ± 0.203
			total labdane derivatives	0.26	20.5	20.6
			iridoid			
1.2	407 ^#^ 361	199 179	antirrhinoside (**3**)	nd	7.58 ± 0.05 ^§^	5.53 ± 0.09 ^§^

## Data Availability

All data generated or analysed during this study are included in this published article.

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
