# Peer review of "NMR, LC-MS Characterization of Rydingia michauxii Extracts, Identification of Natural Products Acting as Modulators of LDLR and PCSK9"

_molecules, 2022, doi:10.3390/molecules27072256_

Round 1
Reviewer 1 Report
Dear Authors,
I believe your manuscript requires extensive correction of the structure elucidation, it's the weakest part of the text. In detail:
Table 1 - standardize the notation of coupling constants and fill in the missing multiplet types;
Table 2 – italicize “R. michauxii”; use “chlorogenic” not “clorogenic”, use “10-deoxycatalpol” not “10 deoxy catalpol”; the table lacks number designations for compounds, particularly for 1-6, that were subjected to in vitro analyses. Reorganize the table so that the compounds are in order of their retention times in the given compound group (hydroxycinnamic/flavonoid/etc.). Compounds at RT 16.1 with m/z of 539 and 493 are probably the same compound – a formic adduct (+46) of 493 (compound 2). An m/z 361 given for 10-deoxycatalpol is also a formic adduct (MW of this compound is 346) and should be noted as [M+FA-H]-.
P 7, L 186 – I don’t understand – is the m/z of compounds with RT>16.1 331, 303, or maybe they all give such fragmentation ions in the MS/MS spectra?
Figure 4 is unnecessary in the main text and should be moved to Supplementary Materials.
The NMR data of compounds 1-3 must be placed in a table with all 1H and 13C NMR chemical shifts and coupling constants. UV-Vis data, optical rotations and ECD data must be delivered for proper structure elucidation and absolute configuration determination - the authors did not indicate the absolute configuration of the compounds studied. D- or L-type of sugar moieties in 2 and 3 must be confirmed after compounds’ hydrolysis. Nevertheless, structure elucidation of 1-2 is seriously flawed – references 31 and 32 given for comp. 1 point to the structure with stereochemistry quite different from the presented in the manuscript (Figure 3) – C-4/C-5 and C-10 are reversed (4R,5R,10R instead of required 4S,5S,10S). Compound 1 with 4R,5R,9R,10R stereochemistry would be a totally new labdane aglycon! You didn’t provide coupling constants for protons H-14 and H-16 (for compounds 1 and 2), nor did you prove that the lactone ring is connected to ethyl bridge through C-13 not C-14. The latter would make compound 2 known - (https://doi.org/10.1016/j.phytochem.2004.01.018 and https://doi.org/10.1016/j.phytochem.2004.06.031). Additionally, the NMR spectra provided in Figures S3-S8 seem not to correspond well with chemical shifts reported in the text, and they don’t support structural elucidation of compound 2 at all.
P9, L 205 – [M-H]- ion of comp. 2 can’t be at m/z 439. Maybe it was 493?
P 9, L 212 – should be “The second ring” not “The second cycle”
P9, L 233 – an utter nonsense – a [M+H]+ at m/z 347 can’t give fragments at m/z 361! Why did you provide [M+H]+? Probably it should [M+FA-H]- at m/z 391. Additionally, the fragmentation pathway of 3 given in Figure S2 is plain wrong – it does not present 10-deoxycatalpol!
Chemical formulae of all isolated compounds (or at least newly isolated) must be supported by measurements of high-resolution mass spectrometry data. NMR data of isolated known compounds (4-6) should be included in the Supplementary Materials in order to be able to confirm their identity.
P10, L257 and P10, L271 – correct the numbering of sections, it should be 2.2. and 2.3, accordingly.
Par. 4.3. – use “Duran” not “Durian”; indicate temperature of NMR measurements
P 15, L 426 – I believe it should be “semipreparative HPLC” not “semipreparative LC-MS” – indicate the HPLC apparatus if applicable, indicate the manufacturer of the HPLC column.
P15, L431 – specify the particle size of the bed and its manufacturer;
P15, L438 and 439 – there is no section “2.3.”, did you mean 4.3.”? You claimed before that “At retention times higher than 16.1 min five intense peaks lacking the UV-VIS spectrum ascribable to phenolic compound were detected”. Why did you use 254 nm for quantification of labdanes? They should rather have UV max absorption at 210 nm.
P15, L440-441 – please describe in detail NMR method used. The provided reference is an auto-citation, and it does not contain a full description of the method used, or conditions used. What was the internal standard?
Whole text – standardize the use of font according to the template, it should be Palatino Linotype; carbohydrate attachment should be written as “glycosylation” not “glycosilation”; the decimal place should be marked with a period “.” rather than a comma “,”.
Author Response
I believe your manuscript requires extensive correction of the structure elucidation, it's the weakest part of the text. In detail:
Table 1 - standardize the notation of coupling constants and fill in the missing multiplet types;
We thank reviewer for suggestion. We implemented the data related NMR signals of Table 1 we included the coupling constant when we could obtain from spectra.
Table 2 – italicize “R. michauxii”; use “chlorogenic” not “clorogenic”, use “10-deoxycatalpol” not “10 deoxy catalpol”; the table lacks number designations for compounds, particularly for 1-6, that were subjected to in vitro analyses. Reorganize the table so that the compounds are in order of their retention times in the given compound group (hydroxycinnamic/flavonoid/etc.). Compounds at RT 16.1 with m/z of 539 and 493 are probably the same compound – a formic adduct (+46) of 493 (compound 2). An m/z 361 given for 10-deoxycatalpol is also a formic adduct (MW of this compound is 346) and should be noted as [M+FA-H]-.
We thank the reviewer for comments, we apologise for mistakes and for incorrect indication in the table 2. We corrected table 2 and all the manuscript with all the request changes. We order the identified compound for each class based on the retention time and we add a reference number to each compound that was tested in in vitro.
We perfectly agree with reviewer related the consideration of formic adducts. We add # and a note in the legend to indicate the formic adducts. We correct the name of 10-deoxycatalpol in all the manuscript because we wrongly give this trivial name while the correct compound is named antirrhinoside and correct the font of figure 3, moving the complete name of compound in the caption.
Related to the antirrhinoside, we apologise for the errors related the mass identification, the [M-H]- ion is at 361 m/z and we observed the formic acid adduct at 407 m/z. The main fragment was observed at 199 and 179 m/z related to the neutral loss of 162 Da. We correct table and all manuscript.
We corrected the table 2 and the related text: “Compound 3 was isolated as an oily residue, the MS spectrum in negative ion mode presented [M+H]- ion at m/z at 361, and the formic acid adduct [M+FA+H]- ion at 407 m/z. Main fragments were observed at m/z 199, 179 suggesting the loss of one hexose unit.”
P 7, L 186 – I don’t understand – is the m/z of compounds with RT>16.1 331, 303, or maybe they all give such fragmentation ions in the MS/MS spectra?
We change in the text the sentence and report the new one: “At retention times higher than 16.1 min four intense peaks lacking the UV-VIS spectrum ascribable to phenolic compound were detected. Three of them resulted in ion species that in MS2 present fragment at m/z 331 in negative ion mode, one showed [M-H]- at m/z 331..”
Figure 4 is unnecessary in the main text and should be moved to Supplementary Materials.
We moved the figure 4 in supplementary material.
The NMR data of compounds 1-3 must be placed in a table with all 1H and 13C NMR chemical shifts and coupling constants.
We thank referee for suggestion, we prepared tables and we placed the table in supplementary materials.
UV-Vis data, optical rotations and ECD data must be delivered for proper structure elucidation and absolute configuration determination - the authors did not indicate the absolute configuration of the compounds studied. D- or L-type of sugar moieties in 2 and 3 must be confirmed after compounds’ hydrolysis.
Thank you very much for your important comment, at present we did not acquired UV -Vis and ECD spectra for the compounds and we have no possibility to acquire EDC spectra in short time so we cannot add this information. We agree on the need for the determination of absolute configuration of sugar and we performed the hydrolisys and identified the sterical serie of the glucose as D so we add this information in the manuscript..
Nevertheless, structure elucidation of 1-2 is seriously flawed – references 31 and 32 given for comp. 1 point to the structure with stereochemistry quite different from the presented in the manuscript (Figure 3) – C-4/C-5 and C-10 are reversed (4R,5R,10R instead of required 4S,5S,10S). Compound 1 with 4R,5R,9R,10R stereochemistry would be a totally new labdane aglycon!
We apologise for our imprecision, and we thank the referee for good suggestion and expertise in NMR analysis. We are grateful for your thorough and insightful revision. During the preparation of the manuscript we wrongly assigned in the formula the absolute configuration of the positions 4, 5, 9 and 10. We checked spectra and we prepared table, and compare spectral data with previously published, also we check optical rotation power and compound structure in our opinion is 4S,5S, 9S,10S derivative. The articles of Zdero (Phytochemistry 1991; 1990) and the Henrick (Tetrahatron 1965) both described similar compounds and in two of the article this derivative.
You didn’t provide coupling constants for protons H-14 and H-16 (for compounds 1 and 2), nor did you prove that the lactone ring is connected to ethyl bridge through C-13 not C-14. The latter would make compound 2 known - (https://doi.org/10.1016/j.phytochem.2004.01.018 and https://doi.org/10.1016/j.phytochem.2004.06.031). Additionally, the NMR spectra provided in Figures S3-S8 seem not to correspond well with chemical shifts reported in the text, and they don’t support structural elucidation of compound 2 at all.
We thank the referee for good comment, the proton signal we assigned to position H-14 is at δ H 5.88 and appear as a singlet. If the side chain was linked at position 14 then this proton will be conjugated with the lactone carbonyl thus presenting a chemical shift at ppm levels around 7.20-7.40. Furthermore, we apologise for our previous description we implemented the details, and we described the correlations that are proving the linkage of the lactone ring to the rest of the molecule. We report the HMBC correlation from the H-14 with C-18. Furthermore, also from H-16 and C-18 we observed HMBC correlations (as hilglighted in the scheme in which HMBC correlation are represented with red arrows). From the H-9 HMBC correlations with C-12 are observed. Thus, in our opinion these correlations support the linkage in position 3. Same correlations were observed in compound 2. In compound 2 that was acquired in deuterated methanol the signal of H16 was under the signal of the OH of methanol.
Compound 1: position-14 δ H 5.88 s, position-16 δH 4.76-4.72 d, J=18.0 Hz in deuterated chloroform
Compound 2: position-14 δ H 5.91 s, position-16 δH 4.88 under the OH of methanol. in deuterated methanol
The 1D and 2D spectra of compounds 1 and 2 in our opinion support the structures of the proposed derivatives. The signals are consistent with the literature in the HSQC of the compound 1 we recorded the presence of two methyl groups, eight aliphatic CH2 one CH2 sp2 one sp2 CH and two sp3 CH. We included spectra in supplementary materials. The compound 2 (with some differences because due to the presence of the sugar residue is not soluble in deuterated chloroform the spectrum was recorded in deuterated methanol) also presents same pattern of signals for the aglycone moiety, with evident signals of a glucopyranosyl unit that due to the observed chemical shift of position 1 (δ 5.45-94.8 J=8.0) suggest the existence of an ester linkage from the acidic group of the ent-labdane and position 1 of glucose. HMBC correlation observed from the H-1of the glucose with carbonyl at 178ppm also support this interpretation. We know that the purity of the compound is not 100% but we started from a little amount of extract thus was not possible to proceed with further purifications.
P9, L 205 – [M-H]- ion of comp. 2 can’t be at m/z 439. Maybe it was 493?
We apologise for mistake, we correct it
P 9, L 212 – should be “The second ring” not “The second cycle”
We apologise for mistake, we correct it
P9, L 233 – an utter nonsense – a [M+H]+ at m/z 347 can’t give fragments at m/z 361! Why did you provide [M+H]+? Probably it should [M+FA-H]- at m/z 391. Additionally, the fragmentation pathway of 3 given in Figure S2 is plain wrong – it does not present 10-deoxycatalpol!
We apologize for mistake, we correct the data related compound 3 in the manuscript and removed incorrect data in supplementary material.
Chemical formulae of all isolated compounds (or at least newly isolated) must be supported by measurements of high-resolution mass spectrometry data. NMR data of isolated known compounds (4-6) should be included in the Supplementary Materials in order to be able to confirm their identity.
We thank for the comment, we agree about the HR-MS but in this moment we cannot provide the data in brief time because we had technical issue with the instrument and we are waiting for maintenance. We think that the MS/MS data can offer sufficient information combined with the NMR. If you absolutely need the HR-MS we can provide the data in few months. About the NMR spectra of the compounds 4-6 we need to reacquire the spectra, because we had issue with hard disk of the machine and we loss some data. Actually we have poor amount of compound, because we used the compounds for biosssays, thus we have dissolved all the isolated amount in DMSO for preparing stock solution to be used for bioassays so if you need NMR for the 4-6 we need some time to re-isolate small amount and acquire the spectra. We thank for all the suggestions and we appreciate all the comments and the suggestions that are increasing the quality of this manuscript. Unfortunately, we performed these separations during the last two years and was complicated to obtain all the needed data.
P10, L257 and P10, L271 – correct the numbering of sections, it should be 2.2. and 2.3, accordingly.
We change number of sections as request
Par. 4.3. – use “Duran” not “Durian”; indicate temperature of NMR measurements
We apologise for mistake, we correct it. The NMR experiment was performed at a temperature of 25 °C.
P 15, L 426 – I believe it should be “semipreparative HPLC” not “semipreparative LC-MS” – indicate the HPLC apparatus if applicable, indicate the manufacturer of the HPLC column.
P15, L431 – specify the particle size of the bed and its manufacturer;
We add the missing information and change sentences as reported: “More hydrophylic fraction pooled on the basis of TLC behaviour was purified by semi-preparative HPLC on Eclipse C18 21.2x 150 mm, 5μm (Agilent Technologies, Santa Clara, CA, USA) semipreparative column using an Agilent 1260 chromatograph (Santa Clara, CA, USA) equipped with 1260 diode array (DAD). Mobile phases were methanol and wa-ter 1% formic acid (30/70) with isocratic gradient, at flow rate of 2,5 mL/min”
P15, L438 and 439 – there is no section “2.3.”, did you mean 4.3.”? You claimed before that “At retention times higher than 16.1 min five intense peaks lacking the UV-VIS spectrum ascribable to phenolic compound were detected”. Why did you use 254 nm for quantification of labdanes? They should rather have UV max absorption at 210 nm.
We apologise for mistakes; we correct it as section 4.3. We quantified ladbane derivative by LC-MS using as reference standard the isolated ent-labda-8(17),13-dien-18-oic acid-15,16-olide. We prepare a standard solution in methanol (100μg/mL) and dilution at concentration of 50, 25,10, 5 ug/ml. MS signal was revealed at 331 m/z and the calibration curve was obtained. We apologize for mistake. Peak do not have UV signal at 254 nm.
P15, L440-441 – please describe in detail NMR method used. The provided reference is an auto-citation, and it does not contain a full description of the method used, or conditions used. What was the internal standard?
We add information about the quantitative NMR in the section. We add a reference (Comai et.al, 2010 Fitoterapia) were the method was illustrated. Caffeine was used as internal standard for quantitative 1H-NMR measurements. A stock solution of caffeine in deuterated chloroform was prepared and 500 μL of this solution (0.5mg/ml) were added to an NMR tube with 10 mg of extract powder.
Whole text – standardize the use of font according to the template, it should be Palatino Linotype; carbohydrate attachment should be written as “glycosylation” not “glycosilation”; the decimal place should be marked with a period “.” rather than a comma “,”.
We apologize for errors in the template and we correct according to suggestions. We change in the text “,” with “.”
Reviewer 2 Report
The manuscript summarised information on NMR, LC-MS characterization of Rydingia michauxii extracts, identification of natural products acting as modulators of LDLR and PCSK9.
Overall, the manuscript looks comprehensive and well summarised. It can be considered for publication after improving its language and grammatical errors.
Author Response
We thank referee for positive comments, we revised all the manuscript and corrected errors
Reviewer 3 Report
The paper presented for review is interesting, however, the discussion should be further elaborated. The study was well-designed, time-consuming and laborious. I would recommend this paper to be accepted after revising carefully.
1. Add the structures of compounds 3, 4 and 5 in the figure 3.
2. In discussion, the results of this article were compared to previous findings. Make the discussion section more detailed.
3. There were many English grammatical mistakes. Please check the one more time.
Author Response
We thank reviewer for the comments, we applied the proposed modification, adding structures and implementing the discussion
Round 2
Reviewer 1 Report
The authors made significant revisions to the manuscript that significantly improved its quality. At this time, I consider the evidence and argumentation presented in the text to be acceptable. Only minor errors need improvement, viz:
The first time you use an abbreviation in the text, present both the spelled-out version and the short form, e.g. NMR, COSY, TOCSY, HSQC, HMBC, DEPT, NOESY, HPLC-DAD-MS, etc.;
Use “Methanol-d4” or “CD3OD” or “MeOD” throughout the Manuscript;
P 9, L 231; P 16, L 478 480 - Use (2R)-butan-2-ol;
P 9, L 249 – correct “HSCQ” to “HSQC”;